

# Estimation of serum C-reactive protein activity in periodontal health and disease and response to treatment: a clinico-biochemical study

Shahabe Saquib Abullais[1], Yogesh Wykole[2], Mohasin Abdul Khader[1], Shaik Mohamed Shamsudeen[3], Sultan Alanazi[4], Shafait Ullah Khateeb[5], Mohammad Yunis Saleem Bhat[6] and Shaheen Shamsuddin[7]

[1] Department of Periodontics, College of Dentistry, King Khalid University, Abha, Saudi Arabia
[2] Consultant periodontist, Mumbai, India
[3] Department of Diagnostic Dental Science, College of Dentistry, King Khalid University, Abha, Saudi Arabia
[4] Department of Preventive Dental Sciences, Faculty of Dentistry, Najran University, Najran, Saudi Arabia
[5] Department of Restorative Dental Science, College of Dentistry, King Khalid University, Abha, Saudi Arabia
[6] Department of Dentistry, Government Medical College and Hospital, Jammu and Kashmir, India
[7] Department of Orthodontics, College of Dentistry, King Khalid University, Abha, Saudi Arabia

Corresponding author
Shahabe Saquib Abullais,
sshahabe@kku.edu.sa

## ABSTRACT

**Background**. Periodontitis is a chronic infectious disease affecting periodontium having multifactorial etiology, can cause significant systemic challengein addition to localized inflammation, tissue damage, and bone resorption. A serological marker of systemic inflammation known as C-reactive protein has been linked to an increased risk for a number of pathological conditions, including cardiovascular diseases.

**Aim**. To estimate levels of serum C-reactive protein in healthy individuals and subjects with periodontal diseases and to compare serum C-reactive protein levels in subjects having periodontal disease pre-operatively & post-operatively.

**Materials and methods**. The study was conducted on 60 subjects age ranging from 35 to 60 years. 30 individuals with healthy periodontium were in group 1 (control group) and the remaining 30 were diagnosed as adult periodontitis were in group 2 (experimental group). Periodontal examination done using gingival index, plaque index, periodontal pocket depth, and Russel's index. CRP levels were examined between group 1 and group 2 and in group 2 between baseline visit before treatment and 2 months after treatment.

**Results**. The findings of this study show a significant connection between periodontal disease and the inflammatory marker CRP in the body, as well as a tendency for a significant decrease in serumCRP levels following periodontitis therapy. At baseline, there was a positive correlation among C-reactive protein, probing pocket depth, and Russell's index.

**Conclusion**. As CRP is a key mediator for cardiovascular disease, an increase in C-reactive protein levels in periodontal diseases suggests a significant connection between periodontitis and cardiovascular diseases. Early periodontal treatment might decrease the severity of cardiovascular disease that already exists. This suggests that periodontal examination should be part of routine practicealong with cardiovascular examination.

# INTRODUCTION

Periodontal disease is a chronic gram negative oral infection which gets initiated in the gingiva, progresses to destruction of alveolar bone and gradual loss of connective tissue supporting the teeth (*Albandar, Brunelle & Kingman, 1999*; *Hajishengallis & Chavakis, 2021*). Recent research indicates that periodontitis patients exhibit greater systemic inflammation, as shown by elevated serum levels of different inflammatory markers as compared to those the control population who are not afflicted by the disease (*Buhlin et al., 2002*; *Noack et al., 2001*; *Botelho, Gao & Jagannathan, 2019*; *Hajishengallis, 2015*). Additionally, these patients have altered lipid profiles (*Ebersole et al., 2017*) that cannot be merely explained by their lifestyle but may be causally linked to endotoxin spread and recurrent episodes of bacteremia (*Katz et al., 2002*; *Bowen et al., 2018*).

As periodontitis advances, changes are observed in host inflammatory mediators (*Iacopino & Cutler, 2000*), localized specific host response and antibacterial response of serum antibody (*Page, 1991*). Bacterial infections frequently provide a strong stimulus for a systemic acute phase response manifested by the increased plasma proteins production including C-reactive protein (CRP) (*Kinane et al., 1993*).

CRP is an acute phase reactant which reacts to a variety of inflammatory stimuli such as heat, injury, infections and hypoxia. CRP level offers a valuable information for the diagnosis, monitoring, and treatment of inflammatory diseases (*McArthur & Clark, 1993*; *Ridker et al., 2000*). CRP levels in serum or plasma rises within 24 to 48 h of acute tissue damage and then decline as trauma or inflammation subsides (*Li & Fang, 2004*). Compared to low density lipid cholesterol, CRP appears to be a stronger predictor of cardiovascular events (*De Ferranti & Rifai, 2007*).

CRP levels are trace in healthy individuals, with values 0.3 mg/L (*Sakkinen et al., 2002*; *Clyne & Olshaker, 1999*). When there is a severe systemic infection, CRP levels might exceed 100 mg/L, which is a helpful measure for tracing the progression of illness (*Kumar et al., 2015*).

Recent epidemiological studies have revealed that individuals with poor periodontal health who are otherwise healthy have higher levels of acute-phase proteins, such as CRP. A high rate of periodontitis should alert doctors to the potential oral cause of an escalating inflammatory load (*Vuong et al., 2020*). Therefore, maintaining oral health may have an impact on overall health, and periodontal therapy will help to achieve just that. This will lower the risk of developing cardiovascular events.

Although poor periodontal health has also been associated with increased systemic serum CRP levels, it is still unknown if effective periodontal therapy can lower CRP levels. Thus, a case control study was conducted to investigate the hypotheses that controlling periodontal inflammation may lower serum CRP levels in view of the aforementioned association between periodontal disease and changes in CRP levels in the serum.

### Aims and objectives

1. To evaluate the serum C-reactive protein levels in both healthy and periodontitis patients.
2. To investigate any possible links between serum C-reactive protein and the known clinical parameters for evaluating gingival and periodontal health.
3. To assess and compare pre- operative and post-operative serum C-reactive protein levels in periodontitis patients.

## MATERIALS & METHODS

The present study was designed as single center, prospective, case control and interventional study which was presented to IRB, College of Dentistry, King Khalid University and got approved with letter reference number (IRB/REG/2020-2021/58). The study involved 60 participants with ages ranging from 35 to 60. Half of the study participants with healthy periodontium were allocated to group 1 (control group) and the remaining 30 participants diagnosed as stage II grade B periodontitis as per the 2017 classification for periodontal disease (*Podzimek et al., 2015*) were assigned to group 2 (experimental group). The diagnosis is based on the clinical examination of patients by a periodontist using a periodontal probe to assess the clinical attachment loss. In addition to that, dental radiographs are also taken to assess the underlying bone condition.

The sample size was calculated by using the mean and standard deviation of two groups from previously published research (*Caton et al., 2018*). The following formula has been used for the estimation of sample size.

$$n = (s_1^2 + s_2^2) \frac{[z_{1-\frac{\alpha}{2}} + z_{1-\beta}]^2}{(\overline{x_1} - \overline{x_2})^2}$$

$$\overline{x_1} = \text{Mean of variable in group1}$$
$$\overline{x_2} = \text{Mean of variable in group2}$$

$s_1$ = standard deviation of variable in group 1
$s_2$ = standard deviation of variable in group 2

$$z_{1-\frac{\alpha}{2}} = \text{Value of nomal deviate at considered level of confidence}$$
$$z_{1-\beta} = \text{Value of nomal deviate at considered power of the study}$$

CPR in group 1; $\overline{x_1} = 2.16$, $s_1 = 0.19$
CPR in group 2; $\overline{x_2} = 2.13$, $s_2 = 0.48$
Entering values in the above given equation yields the following results.

$$n = (0.19)^2 + (0.48)^2 \frac{(1.96 + 0.85)^2}{(2.16 - 2.13)^2}$$

$$n = 20.53 \sim 21$$

We have increased the sample size to 30 to accurate the average values and to increase the validity of the results. Larger sample size also would help to identify outliers in collected data and allow small margins of error. Hence, both groups (test and control) were allotted with 30 samples each.

The inclusion criteria for control group were subjects with clinically healthy gingiva (having bleeding on probing <10%) and patients with moderate periodontitis presenting clinical evidence of periodontal pocket measuring >5 mm around at least six teeth were enrolled in test group. Subjects with history of underlying systemic disease, high cholesterol level, smokers, alcoholics, high body mass index, females who were pregnant or lactating or on hormone replacement therapy and subjects who had received antibiotics or underwent periodontal therapy in past 3 months were excluded from the study.

A special proforma was designed to include systematic and methodological recording of all information and observations. After ethical clearance approval, each participant was given a copy of the study protocol and asked to sign an informed consent form before being included in the study.

### Periodontal examination

The participants' relevant data and various indices were assessed at the first (baseline) visit.

Plaque Index: (Turesky-Gilmore-Glickman modification of the Quigley-Hein Plaque index) (*Turesky, Gilmote & Glickman, 1970*) Assesses plaque on the facial and lingual surfaces of all the teeth after using a disclosing agent. The criteria for scoring were as follows:

Score 0: No plaque

Score 1: Isolated flecks of plaque at the cervical margin

Score 2: A continuous band of plaque up to one mm at the gingival margin

Score 3: Plaque greater than one mm in width and covering up to one third of the tooth surface

Score 4: Plaque covering from one third to two thirds of the tooth surface

Score 5: Plaque covering more than two thirds of the tooth surface

Plaque score was calculated for each individual by adding all scores and dividing by the number of surfaces checked.

### Gingival index (*Loe & Silness, 1963*)

The patients' gingival status is documented using the gingival indices. These indices primarily examine clinical manifestations of inflammation, which are then rated according to predetermined standards. In the Loe and Silness Gingival Index, four surfaces of all teeth (excluded third molars) were scored for the severity of gingivitis. Gingival scoring units were created from the tissues surrounding each tooth: the distal-facial papilla, facial margin, mesio-facial papilla, and the entire lingual gingival margin. The following criteria were used to evaluate each of the four gingival units:

Score 0: Absence of inflammation/normal gingiva.

Score 1: Mild inflammation, slight change in color, slight edema, no bleeding on probing.

Score 2: Moderate inflammation, moderate glazing, redness, edema and hypertrophy, bleeding on probing.

Score 3: Severe inflammation, marked redness and hypertrophy, ulceration and tendency to spontaneous bleeding.

By adding together all of the scores per tooth and dividing by the total number of teeth evaluated, the gingival index score for each individual was calculated.

### *Russell's periodontal index (Russell, 1967)*

The primary indicator of periodontitis is clinical attachment loss. Due to bone loss, it is brought on by the apical migration of junctional epithelium from the cementoenamel junction. Clinical and radiographic evaluation can be used to gauge the extent of bone loss. Additionally indicating the bone loss linked to the tooth are gingival recession and tooth movement. In Russell's Periodontal Index, all the teeth present were examined. The following criteria were used to evaluate the gingival inflammation and periodontal involvement in the entire gingival tissue encircling each tooth (*i.e.,* the entire gingival unit around a tooth).

Score 0: Negative. There is neither overt inflammation in the investing tissues nor loss of function due to destruction of supporting tissue.

Score 1: Mild Gingivitis. An overt area of inflammation in the free gingiva does not circumscribe the tooth.

Score 2: Gingivitis. Inflammation completely circumscribes the tooth, but there is no apparent break in the epithelial attachment.

Score 4: Used only when radiographs are available.

Score 6: Gingivitis with pocket formation. The epithelial attachment has been broken and there is pocket. There is no interference with normal masticatory function, the tooth is firm in its socket and has not drifted.

Score 8: Advanced destruction with loss of masticatory function. The tooth may be loose, may have drifted, may sound dull on percussion with metallic instrument, or may be depressible in its socket.

By adding up all of the individual scores and dividing by the number of teeth examined, the periodontal index score (PI Score) for each person was calculated.

### *Probing pocket depth (PPD)*

An important periodontal finding that identifies the presence of periodontal disease is the depth of the gingival sulcus. To measure the depth of the gingival sulcus, a periodontal probe is inserted into it. This gingival sulcus may be referred to as a periodontal pocket if there is periodontal disease present. The length of the probing depth is measured from the gingival margin to the base of the gingival sulcus. Healthy gingival sulcus usually has probing depth between 1 and three mm. Greater than three mm in depth could be a reason for concern.

The probing pocket depth on mesial, distal, midfacial and mid oral aspects of each tooth was measured with University of Michigan 'o' probe with William's marking (Hu-Friedy Mfg., Chicago, IL, USA). When measuring probing depth, the probe tip was held parallel to the long axis of the tooth and positioned interproximally as closely as possible to the contact point. The probe was gently moved along the root surface towards the bottom of the pocket. The nearest highest millimeter-level measurement was used. By adding together

all the pocket depth readings and dividing that total by the number of surfaces evaluated, the average pocket depth was obtained.

After assessing the periodontal status of the subjects, the following investigations were done.

### Blood sample collection

Blood sample was collected for estimation of serum C-reactive protein from subjects belonging to both groups and two months after periodontal therapy in group 2.

### Estimation of CRP

A quantitative turbidimetric test called CRP-Turbilatex is used for measuring CRP in human plasma or serum. Latex particles coated with specific human anti-CRP form an agglutinating solution when mixed with samples containing CRP. Depending upon the contents of CRP, the agglutination changes the absorbance which can be quantified by comparing with known CRP concentration calibrator.

Test procedure: The automatic analyser was set to wavelength 540 nm and was adjusted to zero absorbance against water. The working reagent (one mL) was mixed with patient sample (5uL) in one test tube and working reagent (one mL) and Calibrator (5uL) in another test tube, and then the absorbance was read immediately (A1) and after two minutes later (A2).

The following is the formula to calculate CRP:

$$\text{mg/L of CRP} = \frac{(A2 - A1)\,\text{Sample}}{(A2 - A1)\,\text{Calibrator}} \times \text{Calibrator concentration}.$$

All the collected experimental data were analyzed using Statistical Package for Social Sciences (SPSS 22.0 Inc., Chicago, IL, USA) software. Data were described as mean and standard deviation; and data analysis was performed to evaluate the statistical difference between groups. Normality test was conducted using the Kolmogorov–Smirnov test and found that the data within each group followed a normal distribution. Paired and unpaired 't' test was used to evaluate the significant difference between the dependent and independent study groups. Relationships between the different parameters used in the current study were tested by Pearson's correlation coefficient. A test $P < 0.05$ was contemplated as statistically significant.

## RESULTS

The age distribution for group 1 and 2 revealed that the mean age for group 1 was 40.73 years and group 2 were 43.26 years. The percentage of female subjects was 33.33% and male subject was 66.66% in group 1, whereas in group 2 percentage of female subjects was 43.33% and male subject was 56.66% (Table 1).

The current study demonstrated different CRP levels in group 1 and 2 at baseline and in group 2 two months after periodontal therapy. The mean serum CRP level in group 1 (subjects with healthy periodontium) at baseline was $0.67 \pm 0.55$ mg/L and in group 2 (subjects having moderate periodontitis) at baseline was $2.34 \pm 0.64$ mg/L at baseline. Two months following periodontal therapy, the mean serum CRP level in group 2 reduced

**Table 1  Demographic data distribution of the subjects.**

| Age & Gender distribution | | Group 1 N (%) | Group 2 N (%) |
|---|---|---|---|
| Age | 35–45 | 21 (70) | 18 (60) |
| | 46–55 | 8 (27) | 4 (13) |
| | Above 55 | 1 (3) | 8 (27) |
| Gender | Male | 20 (67) | 17 (57) |
| | Female | 10 (33) | 13 (43) |

**Table 2  Comparison of plaque index (PI) and gingival index (GI) at baseline for group 1 and group 2.**

| Periodontal indices | Mean ± Standard Deviation | | $t$-value | $p$-value |
|---|---|---|---|---|
| | Group 1 | Group 2 | | |
| Plaque index (PI) | 0.36 ± 0.13 | 2.25 ± 0.24 | 39.16 | <.00001 |
| Gingival Index (GI) | 0.22 ± 0.078 | 2.09 ± 0.163 | 57.4105 | <.00001 |

significantly and it was 1.37 ± 0.42 mg/L. Comparison between the levels of CRP in group 1 and 2 using unpaired $T$-test showed the mean of CRP level was 1.67mg/L with statistically significant $p$ value (<.00001).

The mean baseline values, standard deviation and $p$ values of PI and GI for group 1 and 2 at baseline using unpaired $t$-test showed statistically significant scores (p <.00001) for average PI (0.36 *vs* 2.25) and GI (0.22 Vs 2.09) in group 1 compared to group 2 at baseline (Table 2). The comparison between the levels of CRP for group 2 before and two months after periodontal Therapy using paired $t$-test showed mean of CRP levels in group 2 was 0.97mg/L with statistically significant $p$ value (<.00001).

The mean baseline values, standard deviation, t values and $p$ values of periodontal indices [gingival index (GI), plaque index (PI), probing pocket depth (PPD) and Russell's index (RI)] for group 2 at baseline and two months after periodontal therapy using unpaired $t$-test showed highly statistically significant scores ($p < .00001$) for average probing pocket depth (5.66 *vs* 4.57), plaque index (2.25 *vs* 1.15), gingival index (2.09 *vs* 1.05) and Russell's index (4.56 *vs* 2.26) (Table 3).

Correlation of levels of serum CRP with probing pocket depth (PPD) for group 2 at baseline and two months after periodontal therapy using Pearson's correlation coefficient was 0.51 with a '$p$' value of 0.003 and two months after periodontal therapy it was 0.29 with a '$p$' value of 0.12, which is statistically significant at baseline but not significant after two months of periodontal therapy (Table 4, Fig. 1). Correlation of serum CRP levels with plaque index (PI) for group 2 at baseline and two months after periodontal therapy using Pearson's correlation coefficient was −0.042 with a '$p$' value 0.82 and two months after periodontal therapy it was −0.049 with a '$p$' value 0.79, which is statistically not significant at baseline and two months after periodontal therapy (Table 4, Fig. 2).

Correlation of serum CRP levels with GI for group 2 at baseline and two months after periodontal therapy using Pearson's correlation coefficient at baseline was −0.062 with a '$p$' value 0.74 and two months after periodontal therapy it was −0.20 with a '$p$' value

**Table 3** Comparison of various periodontal indices for group 2 at baseline and two months after periodontal therapy.

| Periodontal indices | Mean ± Standard Deviation | | $t$-value | $p$-value |
|---|---|---|---|---|
| | **At baseline** | **After 2 months** | | |
| Probing pocket depth (PPD) | 5.66 ± 0.488 | 4.57 ± 0.53 | 9.571 | <.00001 |
| Plaque index (PI) | 2.25 ± 0.24 | 1.15 ± 0.21 | 18.633 | <.00001 |
| Gingival index (GI) | 2.09 ± 0.163 | 1.05 ± 0.24 | 28.4777 | <.00001 |
| Russel's index (RI) | 4.56 ± 1.23 | 2.26 ± 1.43 | 15.80 | <.00001 |

**Table 4** Correlation of levels of serum C-reactive protein (CRP) with periodontal indices for group 2 at baseline and two months after periodontal therapy.

| Periodontal indices | Pearson's correlation coefficient ($r$) | | $p$-value |
|---|---|---|---|
| Probing pocket depth (PPD) | At baseline | 0.51 | 0.003 |
| | After 2 months | 0.29 | 0.12 |
| Plaque index (PI) | At baseline | −0.042 | 0.82 |
| | After 2 months | −0.049 | 0.79 |
| Gingival index (GI) | At baseline | −0.062 | 0.74 |
| | After 2 months | −0.20 | 0.28 |
| Russel's index (RI) | At baseline | 0.44 | 0.01 |
| | After 2 months | 0.27 | 0.14 |

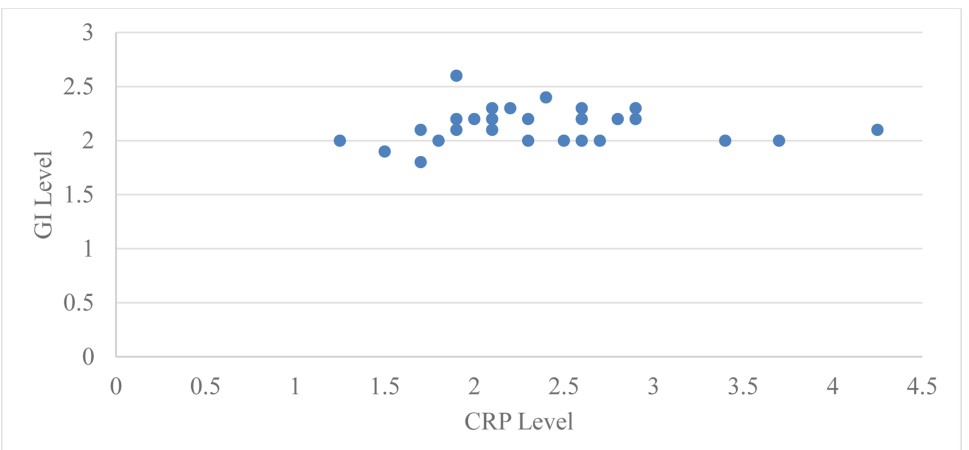

**Figure 1** Relation between CRP level and GI level at baseline in group 2.

of 0.28, which is statistically not significant at baseline and two months after periodontal therapy (Table 4, Fig. 3).

Pearson's correlation coefficient was used to assess the correlation between serum CRP levels and Russell's index (RI) in group 2 (subjects with moderate periodontitis) at baseline and two months following periodontal treatment. The Pearson's correlation coefficient for the subjects in group 2 was 0.44 at baseline with a '$p$' value of 0.01 and 0.27 at two months

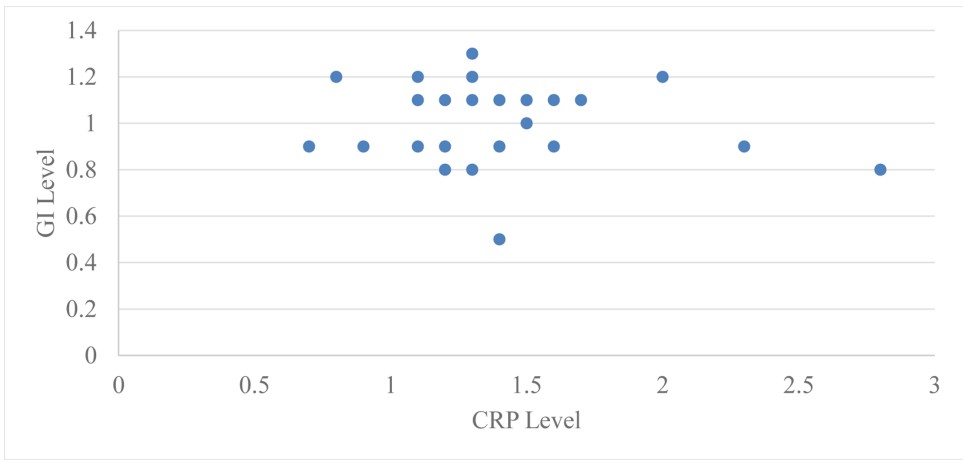

**Figure 2** Relation between CRP level and GI level after 2 months in group 2.

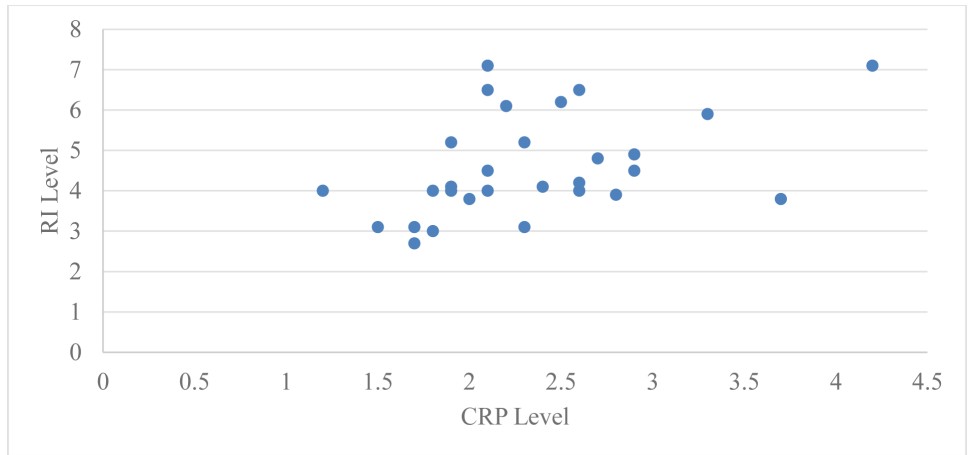

**Figure 3** Relation between CRP level and RI level at baseline in group 2.

with a '$p$' value of 0.14, which is statistically significant at baseline but not significant at two months after periodontal therapy (Table 4, Fig. 4).

## DISCUSSION

Periodontal disease having a multifactorial etiology, represents a previously unrecognized risk factor for atherosclerosis and thromboembolic events. Once periodontal disease is established, lipopolysaccharides and inflammatory cytokines, particularly Interleukin (IL)-1$\beta$, Prostaglandin E2 (PGE2) and Tumor Necrosis Factor (TNF)-$\alpha$ enhance a biologic burden which initiates the onset and progression of atherogenesis and thromboembolic events (*Slade et al., 2000*).

In case of periodontal inflammation, the endotoxin of gram-negative microorganisms interacts with toll-like receptors (TLR), which is expressed on the surface of neutrophils

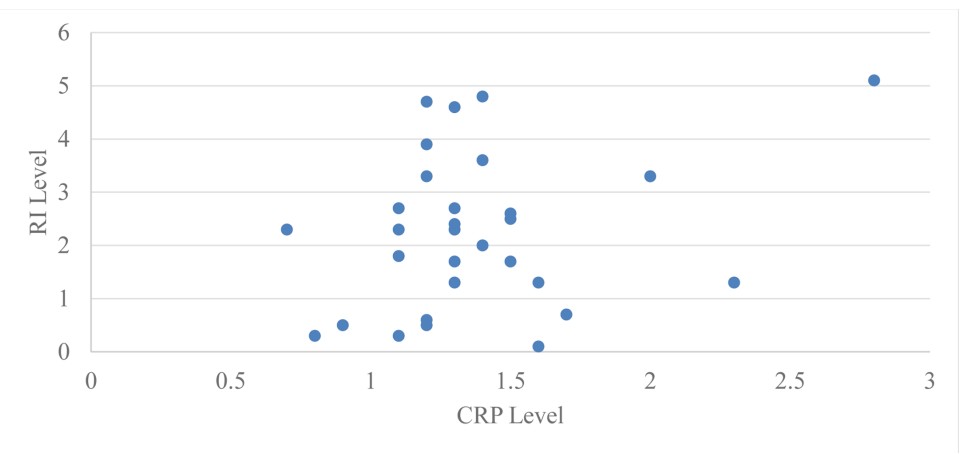

**Figure 4** Relation between CRP level and GI level after 2 months in group 2.

and monocytes. In both the innate and adaptive immune systems, TLR-ligand complexes trigger signal transduction pathways that result in the production of cytokines, which regulate both the local and systemic inflammatory response (*Akira & Takeda, 2004*). Hepatocytes are activated by pro-inflammatory cytokines produced at the site of local disease to produce acute phase proteins, including CRP, which is a component of the non-specific response (*Bansal, Pandey & Asthana, 2014*). Epidemiological research has revealed that systemically healthy individuals with poor periodontal status had higher levels of acute phase proteins, such as CRP (*Bansal, Pandey & Asthana, 2014*).

CRP is a serological marker of systemic inflammation that has been linked to a higher risk of a number of systemic illnesses, such as cardiovascular diseases and unfavorable pregnancy outcomes (*Wu, Trevisan & Genco, 2000*; *Pitiphot, 2005*). Additionally, increased CRP levels in adults and pregnant women have been associated to periodontitis. Therefore, it is hypothesized that CRP may act as a potential mediator in the relationship between periodontitis and these systemic diseases.

The results of the present study revealed that in test group CRP levels were more elevated than those of control group. The results of our research are comparable to those of *Yamazaki et al. (2005b)* and *Pitiphat, Savetsilp & Wara-Aswapati (2008)*. This suggests that periodontitis in these otherwise healthy individuals increases their systemic inflammatory load. The release of inflammatory mediators (IL-1, 1L-6) in systemic circulation and distant inflammatory response induced by periodontal pathogens are among the proposed mechanistic reasons (*Gani et al., 2009*).

Two months after periodontal therapy, there was a statistically significant reduction in serum CRP levels in test group. The results obtained in present study are consistent with the results obtained by *Lalla et al. (2007)*, demonstrated a reduction in levels of serum CRP in moderate to severe periodontitis patients. The results of this study agree with those of *D'Aiiito, Ready & Tonetti (2004)* analyzed the effects of non-surgical periodontal treatment on serum CRP and reported a significant reduction in serum CRP levels at two and six post-operative months. This contrasts with the results of *Ebersole et al. (1997)* didn't

observe a reduction in circulating CRP after nonsurgical periodontal treatment. However, they reported a reduction if an anti-inflammatory drug was administered.

In the presnt study, the mean CRP level was reduced by 58.25%. CRP levels decreased by 59% in those whose baseline CRP was >2 mg/L and by an average of 56% in those whose baseline CRP was <2 mg/L, respectively. Thus, it implies that periodontal infection may contribute to systemic inflammation and a course of non-surgical treatment of moderate periodontitis did influence circulating serum levels of CRP.

The level of CRP in serum indicates the likelihood of a future cardiovascular disease. A person has a minimal risk of getting cardiovascular disease if their high sensitivity (hs)-CRP level is <1 mg/L. A person has an average risk when their hs-CRP level is between 1.0 and 3.0 mg/L and a high risk when it is >3 mg/L (*Zhang et al., 2022*).

In the present study two months post-operative results revealed that only three participants had CRP levels below 1.0 mg/L, leaving 27 subjects with a moderate risk of developing cardiovascular disease. While recruiting subjects, we made an effort to rule out as many potentially confounding factors as we could. However, possibly some study participants did have conditions other than periodontitis that could have affected the levels of acute-phase proteins and might not have responded to periodontal treatment. An alternative explanation can be, a standard non-surgical periodontal therapy was given to 30 subjects in test group. The levels of disease observed in these participants and the concomitant decrease in inflammatory and infectious load would not have been sufficient to cause appreciable alterations in CRP.

In test group, the correlation between CRP and PPD at baseline and after periodontal therapy showed that both were statistically significant only at baseline ($p = 0.003$) but not after two months of periodontal therapy ($p = 0.12$). These findings are similar to those of the research by *Yamazaki et al. (2005a)* that evaluated whether the presence of chronic periodontitis and subsequent periodontal therapy may affect CRP levels in the serum.

In the current study, average PI at baseline in test group was 2.25 which significantly reduced to 1.15 two months after periodontal therapy. There was no correlation between PI and serum CRP levels at baseline ($p = 0.82$) and two months after periodontal therapy ($p = 0.79$). These results are in accordance with those of *Ide et al. (2003)*, performed research to evaluate effect of periodontal disease therapy on circulating levels of inflammatory and cardiovascular markers. Research by *Mattila et al. (2002)* aimed to determine whether treating periodontitis may reduce the CRP and fibrinogen levels. They concluded that neither the amount of the reduced nor the elevated baseline CRP levels were related to the severity of periodontitis.

According to the findings of our study, there was no correlation between the severity of the plaque score at baseline and the levels of hs-CRP or two months after periodontal therapy was completed.

In our study, gingival index (GI) and serum CRP levels did not correlate either at baseline ($p = 0.74$), or two months after periodontal treatment ($p = 0.28$). These results are in line with those of *Mattila et al. (2002)* and *Ide et al. (2003)*. According to our study, there was no correlation between the GI score's severity and hs-CRP level at baseline and two months after periodontal therapy. Russell's index in test group was 4.56 at baseline

and significantly decreased to 2.26 two months after periodontal treatment. In test group, the correlation between CRP and RI at baseline and two months following periodontal treatment revealed that both were only statistically significant at baseline. Both parameters showed improvement after periodontal therapy but correlation could not be established.

Individuals with moderate periodontitis had higher serum CRP levels than those with healthy periodontium. Two months after periodontal treatment, patients with moderate periodontitis showed a significant reduction in serum CRP level. Serum CRP levels and PPD and RI showed a positive correlation at baseline, but not two months after periodontal treatment. Serum CRP, PI, and GI were neither correlated at baseline nor two months after periodontal treatment.

The outcome of the periodontal therapy is also impacted by the fixed prosthesis. Patients with inflammatory periodontal tissues exhibit signs of inflammation. The fixed prosthesis might make the periodontal condition worse if the oral hygiene status is not preserved by the elimination of plaque (*Avetisyan et al., 2021*). The appropriate prosthesis for patients with periodontal diseases might be chosen using a cytomorphometric study. Changes in the GCF's composition might be used as a valuable factor when deciding the prosthesis (*Heboyan et al., 2020*).

The following limitations require thoughtful consideration while conducting further studies in relation to similar problems. Screening of a larger sample size is desirable. In the present study, 60 subjects were evaluated and only 30 were treated. The current study employed a follow-up period of two months in test group. The link between serum CRP levels and periodontal therapy will be better understood with the use of long-term clinical observations and prospective research. In this study a standard non-surgical treatment was given to subject having moderate periodontitis. It would be more intriguing to find out the effect of surgical periodontal therapy and other adjunct therapy on the level of serum CRP.

## CONCLUSIONS

In the current study, patients with moderate periodontitis had higher serum CRP levels as compared to patients with clinical periodontal health. Patients with moderate periodontitis showed a reduction in serum CRP level followed by periodontal therapy. No correlation was reported among serum CRP, PI, and GI at baseline or two months post periodontal therapy.

An increase in serum CRP is a risk factor for cardiovascular disease. Early detection and treatment of periodontitis can reduce the risk for cardiovascular disease as compared to otherwise healthy people. Additionally, periodontal therapy may reduce the severity of existing cardiovascular disease. This suggests that periodontal status evaluation should be a part of routine cardiovascular examination.

### Funding

This work was supported by the Deanship of Scientific Research at King Khalid University through Large group Project RGP-2/255/44. The funders had no role in study design, data collection and analysis, decision to publish, or preparation of the manuscript.

### Grant Disclosures

The following grant information was disclosed by the authors:
Deanship of Scientific Research at King Khalid University through Large group Project: RGP-2/255/44.

### Competing Interests

The authors declare there are no competing interests.

### Author Contributions

- Shahabe Saquib Abullais conceived and designed the experiments, prepared figures and/or tables, and approved the final draft.
- Yogesh Wykole conceived and designed the experiments, prepared figures and/or tables, and approved the final draft.
- Mohasin Abdul Khader performed the experiments, prepared figures and/or tables, and approved the final draft.
- Shaik Mohamed Shamsudeen performed the experiments, prepared figures and/or tables, and approved the final draft.
- Sultan Alanazi analyzed the data, authored or reviewed drafts of the article, and approved the final draft.
- Shafait Ullah Khateeb analyzed the data, authored or reviewed drafts of the article, and approved the final draft.
- Mohammad Yunis Saleem Bhat analyzed the data, authored or reviewed drafts of the article, and approved the final draft.
- Shaheen Shamsuddin performed the experiments, authored or reviewed drafts of the article, and approved the final draft.

### Human Ethics

The following information was supplied relating to ethical approvals (*i.e.*, approving body and any reference numbers):

The IRB of the College of Dentistry, King Khalid University granted Ethical approval to carry out the study within its facilities.

### Data Availability

The raw data are available in the Supplemental Files.

## Supplemental Information

Supplemental information for this article can be found online at http://dx.doi.org/10.7717/peerj.16495#supplemental-information.

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
