# Peer review of "Estimation of serum C-reactive protein activity in periodontal health and disease and response to treatment: a clinico-biochemical study"

_PeerJ, doi:10.7717/peerj.16495_

## Round 0.1 · original submission · Major Revisions

Thanks for your submission. Please address the following queries-

1. It needs to be mentioned what type of study design was used.

2. How the sample size was determined? How did you allocate an equal number of participants to each group?

3. What are the inclusion criteria for selecting a participant in this study?

4. There is not much information of key variables in the study.

5. What type of analyses were performed? Which statistical software was used?

6. Did you assess the normality? No information on normality checking and you performed t- test. Needed clarification.

·

Basic reporting

This is an interesting study where the researchers compared the levels of serum C-reactive protein in healthy individuals and subjects with periodontal diseases.

Abstract
The aim is long and repetition of the words. Please edit.
Please write the full form of the abbreviation when using it for the first time.

Experimental design

How the sample size calculation was done. Please explain more.

There is no Statistical analysis part. Please add this section.

Validity of the findings

Please edit the conclusion and present a better conclusion relevant to this study.

Additional comments

It seems patients with fixed prosthetic treatment are included in this study. The fixed prosthetic treatment has an effect on the periodontal outcome. Add more discussion on periodontal health and CRP in fixed prosthetic treatment.
https://pubmed.ncbi.nlm.nih.gov/33801337/
https://www.ncbi.nlm.nih.gov/pmc/articles/PMC7587339/

The summary section should be removed but can be discussed in the Discussion.

Reviewer 2 ·

Basic reporting

A bit of language editing is required for further processing.
All mentioned literatures are older, there are only a few may be 4/5 new literature from 2015-2023. There was no mention of newer periodontal disease classification (2017) though the study was probably conducted in 2020-2021.
The authors have selected a periodontitis case away from the inclusion criteria, that they have mentioned.

Experimental design

Good

Validity of the findings

Results are presented in a vague manner. It is therefore recommended to re-write, and make it concise and attractive before resubmission.

Additional comments

Please download 1 sample published by Peer J recently and prepare your manuscript. The definition of periodontitis as per newer guidelines has to be used and be strict to your inclusion criteria. Your raw data suggests that you have taken Pocket depth >5mm as a periodontitis patient but you have mentioned it 2-5mm will be your candidates in the inclusion criteria. BMI was taken and no mention of why it was taken was quite intriguing. There are a few spelling corrections in line number 36,135. Please provide your proforma sheet and institutional review committee paper as well.

Annotated reviews are not available for download in order to protect the identity of reviewers who chose to remain anonymous.

---

## Round 0.2 · Minor Revisions

Dear authors, It is crucial to ensure the quality and validity of the research presented in your paper, and I would like to request your attention to the following issues:

Statistical Analysis: I have reservations regarding the explanation of how you conducted the statistical analysis. Before performing a t-test, it is advisable to assess normality using tests like the Shapiro-Wilk or Kolmogorov-Smirnov tests to check if the data in each group follows a normal distribution. If the data in one or both groups significantly deviates from a normal distribution, it may be necessary to consider non-parametric tests, such as the Wilcoxon-Mann-Whitney U test, instead of a t-test. In summary, please check for normality in each group. If both groups follow a normal distribution and have approximately equal variances, you can proceed with an unpaired t-test. If normality is not met or variances differ significantly, consider using non-parametric tests or other appropriate statistical methods. - these should be however clear in your work description!

Diagnosis Method: It appears that there is a missing description of how the diagnosis was performed in your manuscript. It would be valuable to include figures or data related to the diagnostic process, whether it involves biochemistry or pathology. This additional information will enhance the clarity and comprehensibility of your work.

P-Values: I also noticed that p-values of 0.000 were reported in Tables 2 and 3. I have concerns about the accuracy of these values, as it is more likely that they are simply lower than the program's display threshold. It is essential to clarify this point to ensure the correctness of your statistical reporting.

I believe it is crucial to address these issues to enhance the quality and validity of your research before it is accepted for publication.

I look forward to your response and the necessary revisions to address these concerns.

---

## Round 0.3 · accepted · Accept

Dear authors, thank you for completing the requested revisions. I am happy to let you know that your manuscript has been accepted for publication in PeerJ.